# ADVERSARIAL PARTIAL MULTI-LABEL LEARNING

## ABSTRACT

Partial multi-label learning (PML), which tackles the problem of learning multi-label prediction models from instances with overcomplete noisy annotations, has recently started gaining attention from the research community. In this paper, we propose a novel adversarial learning model, PML-GAN, under a generalized encoder-decoder framework for partial multi-label learning. The PML-GAN model uses a disambiguation network to identify noisy labels and uses a multi-label prediction network to map the training instances to the disambiguated label vectors, while deploying a generative adversarial network as an inverse mapping from label vectors to data samples in the input feature space. The learning of the overall model corresponds to a minimax adversarial game, which enhances the correspondence of input features with the output labels. Extensive experiments are conducted on multiple datasets, while the proposed model demonstrates the state-of-the-art performance for partial multi-label learning.

## 1 INTRODUCTION

In partial multi-label learning (PML), each training instance is assigned multiple candidate labels which are only partially relevant; that is, some assigned labels are irrelevant noise. As it is typically difficult and costly to precisely annotate instances for multi-label data (Xie & Huang, 2018), the task of PML naturally arises in many real-world scenarios with crowdsource annotations. Figure 1 presents an example of training images for object recognition under the PML setting, where the union of candidate labels provided by crowdsource annotators is overcomplete and contains both ground truth labels (in black color) and irrelevant noise labels (in red color). PML is much more challenging than standard multi-label learning as the true labels are hidden among irrelevant labels and the number of true labels is unknown. The goal of PML is to learn a good multi-label prediction model from such a partial label training set, and hence reduce the annotation cost.

An intuitive strategy of PML is to treat all candidate labels as relevant ground truth, thus any off-the-shelf multi-label classification methods can be adapted to induce an expected multi-label predictor (Zhang & Zhou, 2014). This strategy, though simple, cannot work well since taking the noisy labels as part of the true labels will mislead the multi-label training and induce inferior prediction models. The PML work in (Xie & Huang, 2018) assumes that each candidate label has a confidence score of being a true label, and learns the confidence scores and classifier in an alternative manner by minimizing a confidence weighted ranking loss between the candidate and non-candidate labels. Although this work yields some reasonable results, the estimation of label confidence scores is error-prone, especially when noisy labels dominate, which can seriously impair the classifier's performance. Another recent work in (Fang & Zhang, 2019) also exploits label confidence values of candidate labels for PML. It estimates the confidence values using iterative label propagation and chooses the candidate labels with high confidence values as credible labels, which are then used to induce a multi-label prediction model. This work however suffers from the cumulative errors induced in propagation, which can impact the estimation of the credible labels and consequently impair the prediction model.

In this paper, we propose a novel adversarial learning model, PML-GAN, under a generalized encoder-decoder framework to tackle the partial multi-label learning problem. The PML-GAN model comprises four component networks: a disambiguation network that predicts the probability of each candidate label being an additive noise for a training instance; a prediction network that predicts the disambiguated *true* labels of each instance from its input features; a generation network that generates samples in the feature space given latent vectors in the label space; and a discrimination network that separates the generated samples from the real data. The prediction network and

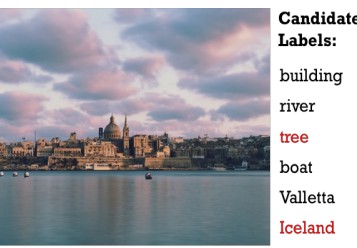

Figure 1: An annotated image under the partial multi-label learning (PML) setting.

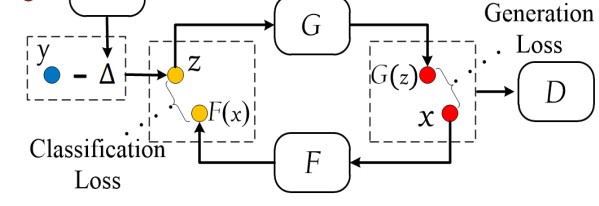

Figure 2: The proposed PML-GAN model. It has four component networks: generator $G$, disambiguator $\widetilde{D}$, predictor $F$, and discriminator $D$.

disambiguation network together form an encoder that maps data samples in the input feature space to the disambiguated label vectors, while the generation network and discrimination network forms a generative adversarial network (GAN) (Goodfellow et al., 2014) as an inverse decoding mapping from vectors in the multi-label space to samples in the input feature space. The learning of the overall model corresponds to a minimax adversarial game, which enhances the correspondence of input features with the output labels, and consequently boosts multi-label prediction performance. To the best of our knowledge, this is the first work that exploits GANs for PML. We conduct extensive experiments on multi-label datasets under partial multi-label learning setting. The empirical results show the proposed PML-GAN yields the state-of-the-art PML performance.

## 2 RELATED WORK

Multi-label learning is a prevalent classification problem in many real world domains, where each instance can be assigned into multiple classes simultaneously. Many multi-label learning methods developed in the literature exploit label correlations at different degrees to produce multi-label classifiers (Zhang & Zhou, 2014), including the first order methods (Zhang et al., 2018), second order methods (Li et al., 2014), and high-order methods (Burkhardt & Kramer, 2018). Nevertheless, standard multi-label learning methods all assume each training instance is annotated with a complete set of ground truth labels, which can be impractical in many domains, where the annotations are obtained through crowdsourcing. With the union of annotations produced by multiple noisy labelers under the crowdsourcing setting, the partial multi-label learning (PML) problem arises naturally in real world scenarios, where the set of labels assigned to each training instance not only contain the ground truth labels, but also some additional irrelevant labels.

PML is more challenging than standard multi-label learning. The previous PML work in (Xie & Huang, 2018) proposes two methods, PML-FP and PML-LC, to estimate the label confidence values and optimize the relevance ordering of labels on each training instance by exploring the structural information in both feature and label spaces. However, due to the inherent property of alternative optimization, in these methods, the estimation error of labeling confidence values can negatively impact the coupled multi-label predictor. The work in (Sun et al., 2019) denoises the observed label matrix based on low-rank and sparse matrix decomposition. In another recent work (Fang & Zhang, 2019), the authors propose to address PML problem using a two-stage strategy. It first estimates the label confidence value of each candidate label with iterative label propagation, and then performs multi-label learning over selected credible labels based on the confidence values by using pairwise label ranking (PARTICLE-VLS) or maximum a posteriori reasoning (PARTICLE-MAP). However, their credible label estimation can be impaired by the cumulative error induced in the propagation process, which can consequently degrade the multi-label learning performance, especially when there are many noisy labels.

Studies on weak learning and partial label learning have some connections with PML, but address different problems. Weak label learning tackles the problem of multi-label learning with incomplete labels (Sun et al., 2010; Wei et al., 2018), where some ground truth labels are missed out from the annotations. Partial label learning (PLL) tackles multi-class classification under the setting where for each training instance there is one ground-truth label among the given candidate label set (Cour et al., 2011; Liu & Dieterich, 2012; Zhang & Yu, 2015; Yu & Zhang, 2016; Chen et al., 2018). PLL

methods cannot be directly applied on the more challenging PML problems, as under PML one has unknown numbers of ground truth labels among the candidate label set for each training instance.

Generative adversarial networks (GANs) (Goodfellow et al., 2014), which perform minimax adversarial training over a generation network and a discrimination network, are one of the most popular generative models since its introduction. During the past years, a vast range of GAN-based adversarial learning methods have been developed to address different tasks, including semi-supervised learning (Kumar et al., 2017; Lecouat et al., 2018), unsupervised learning (Jakab et al., 2018), and learning with noisy labels (Thekumparampil et al., 2018). The proposed work in this paper however is the first one that exploits GAN models for PML.

## 3 PROPOSED APPROACH

In this section, we present the proposed adversarial partial multi-label learning model, PML-GAN, under the following setting. Assume we have a training set $S = (X, Y) = \{(\mathbf{x}_i, \mathbf{y}_i)\}_{i=1}^{N}$, where $\mathbf{x}_i \in \mathbb{R}^d$ denotes the input feature vector for the $i$-th instance, and $\mathbf{y}_i \in \{0, 1\}^L$ is the corresponding annotated label indicator vector. The multiple 1 values in each $\mathbf{y}_i$ indicate either the ground truth labels or the additional mis-annotated labels. We aim to learn a good multi-label prediction model from this partially labeled training set.

The proposed PML-GAN model is illustrated in Figure 2, which comprises four component networks, disambiguation network $\widetilde{D}$, prediction network $F$, generation network $G$ and discrimination network $D$. The four components coordinate with and enhance each other under an encoder-decoder learning framework, which forms inverse mappings between the instance vectors in the input feature space and the label vectors in the output class label space. Below we present these model components, the learning objective and training algorithm in details.

### 3.1 PREDICTION WITH DISAMBIGUATED LABELS

Comparing to standard multi-label learning, the main difficulty of PML is that the annotated labels $\{\mathbf{y}_i\}$ in the training data contain additive noisy labels. The main challenge lies in identifying the ground truth labels $\mathbf{z}_i^*$ from each annotated candidate label vectors $\mathbf{y}_i$; that is dropping the additional 1s from each candidate label vector $\mathbf{y}_i$. We propose to tackle this challenge by using a disambiguation network $\widetilde{D} : \Omega_{\mathbf{x}} \rightarrow \Omega_{\Delta}$ ($\Omega_{\cdot}$ denotes the corresponding domain space), which predicts the irrelevant labels for a given instance. Hence the true label vector $\mathbf{z}_i^*$ can be recovered as $\mathbf{z}_i^* = \text{ReLU}(\mathbf{y}_i - \Delta_i)$, where $\text{ReLU}(\cdot) = \max(\cdot, 0)$ denotes the commonly used rectified linear unit activation function for deep neural networks. Then we can learn a prediction network $F : \Omega_{\mathbf{x}} \rightarrow \Omega_{\mathbf{z}}$, i.e., a multi-label classifier, to predict the disambiguated ground truth labels for each instance.

Although the label indicator vectors in the training data are provided as discrete values, it is difficult for either the disambiguation network or the prediction network to directly produce discrete output values. Instead, by using a sigmoid activation function on the last layer of each network, $\widetilde{D}(\mathbf{x})$ and $F(\mathbf{x})$ can predict the probability of each class label being the additive irrelevant label and the ground truth label respectively. With the disambiguation network and prediction network, we can perform partial multi-label learning by minimizing the *classification loss* on training data $S$:

$$\min_{F, \widetilde{D}} \quad \mathcal{L}_c(X, Y; F, \widetilde{D}) = \sum_{(\mathbf{x}_i, \mathbf{y}_i) \sim S} \ell_c(F(\mathbf{x}_i), \mathbf{z}_i) \tag{1}$$

$$\text{s.t.} \quad \mathbf{z}_i = \text{ReLU}(\mathbf{y}_i - \Delta_i), \quad \Delta_i = \widetilde{D}(\mathbf{x}_i), \quad \forall(\mathbf{x}_i, \mathbf{y}_i) \sim S$$

where $\mathbf{z}_i$ denotes the disambiguated label confidence vector with continuous values in $[0, 1]$, which can be viewed as a relaxation of a true label indicator vector, while $\ell_c(\cdot, \cdot)$ denotes the cross-entropy loss between the predicted probability of each label and its confidence of being a ground-truth label.

### 3.2 INVERSE MAPPING WITH GANS

The prediction network and disambiguation network together form an encoder that maps data samples in the input feature space to the disambiguated label vectors. To enhance the label disambiguation and hence improve multi-label classification, we propose to conduct an inverse decoding mapping

from label vectors $\hat{\mathbf{z}} \in \{0, 1\}^L$ to samples in the input feature space. In particular, we propose to deploy a generative adversarial network (GAN) model to generate samples in the input feature space given label vectors in the label space. The GAN model comprises a generation network $G$ and a discrimination network $D$. Given a label vector $\hat{\mathbf{z}}$ sampled from a prior binomial distribution $P(\hat{\mathbf{z}})$, one can generate a sample $\hat{\mathbf{x}}$ using the generation network, $\mathbf{x} = G(\hat{\mathbf{z}})$. A two-class discriminator $D$ is used to discriminate the generated samples from the real samples in $S$. The training of the GAN model is a minimax optimization problem over an *adversarial loss* function:

$$\min_G \max_D \quad \mathcal{L}_{adv}(G, D, S) = \mathbb{E}_{\mathbf{x}_i \sim S}[\log D(\mathbf{x}_i)] + \mathbb{E}_{\hat{\mathbf{z}} \sim P(\hat{\mathbf{z}})}[\log(1 - D(G(\hat{\mathbf{z}})))] \tag{2}$$

where the discriminator $D$ tries to maximally distinguish the generated samples $G(\hat{\mathbf{z}})$ from the real data samples in $S$, and the generator $G$ tries to generate samples that are similar to the real data as much as possible such that the discriminator cannot tell the difference.

In theory, the samples generated by the adversarially trained generator $G$ can have an identical distribution with the real data $S$ (Goodfellow et al., 2014). But this does not guarantee the generated samples can match the real training samples. To ensure the generator $G$ can provide an inverse mapping function relative to the predictor $F$, we further propose to decode the disambiguated training label vectors into the training samples $S$ with $G$ by deploying a *generation loss*:

$$\mathcal{L}_g(G, S) = \sum_{(\mathbf{x}_i, \mathbf{y}_i) \sim S} \ell_g(G(\mathbf{z}_i), \mathbf{x}_i), \quad \text{where} \quad \mathbf{z}_i = \text{ReLU}(\mathbf{y}_i - \widetilde{D}(\mathbf{x}_i)) \tag{3}$$

where $\ell_g(\cdot, \cdot)$ measures the generation loss on each training instance and can be a least squares function. This generation loss can enhance the label disambiguation and improve multi-label learning.

### 3.3 LEARNING WITH PML-GANS

By integrating the classification loss in Eq.(1), the adversarial loss in Eq.(2), and the generation loss in Eq.(3) together, we obtain the following minimax optimization problem for the proposed PML-GAN model:

$$\min_{G, \widetilde{D}, F} \max_D \quad \mathbb{E}_{(\mathbf{x}_i, \mathbf{y}_i) \sim S}\Big(\ell_c(F(\mathbf{x}_i), \mathbf{z}_i) + \ell_g(G(\mathbf{z}_i), \mathbf{x}_i)\Big) +$$

$$\beta \Big(\mathbb{E}_{\mathbf{x}_i \sim S}[\log D(\mathbf{x}_i)] + \mathbb{E}_{\hat{\mathbf{z}} \sim P(\hat{\mathbf{z}})}[\log(1 - D(G(\hat{\mathbf{z}})))]\Big) \tag{4}$$

$$\text{s.t.} \quad \mathbf{z}_i = \text{ReLU}(\mathbf{y}_i - \widetilde{D}(\mathbf{x}_i)), \quad \forall(\mathbf{x}_i, \mathbf{y}_i) \sim S$$

where $\beta$ is a trade-off hyperparameter that controls the relative importance of the generation loss and adversarial loss respectively; the objective function can be denoted as $\mathcal{L}(G, \widetilde{D}, F, D)$. The learning of the overall model corresponds to a minimax adversarial game, which enhances the correspondence of input features with the output labels in an inverse encoder-decoder manner, and consequently boosts multi-label prediction performance.

We perform training using a minibatch based stochastic gradient descent algorithm. In each iteration of the training, the minimization over $G, \widetilde{D}, F$ and the maximization over $D$ are conducted alternatively. The overall training algorithm is presented in Algorithm 1.

## 4 THEORETICAL RESULTS

In the proposed PML-GAN model, given the generator $G$, the discriminator $D$ is conditionally independent from the predictor $F$ and disambiguator $\widetilde{D}$. Between $G, F$ and $\widetilde{D}$, $G$ and $F$ are conditionally independent from each other given $\widetilde{D}$. Their independence relationship can be illustrated using the undirected dependence graph in Figure 3. Based on these conditional independence relationships, we have the following optimality results.

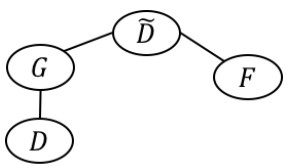

Figure 3: Dependence graph of PML-GAN.

---

**Algorithm 1** Minibatch stochastic gradient descent training of PML-GAN

---

**Input**: training set $S$; trade-off parameter $\beta$; $k$– # of update steps for the discriminator.

**for** number of training iterations **do**

   Sample a minibatch of $m$ samples $\{(\mathbf{x}_1, \mathbf{y}_i), \cdots, (\mathbf{x}_m, \mathbf{y}_m)\}$ from training set $S$.

   Sample $n$ label vectors $\{\hat{\mathbf{z}}_1, \cdots, \hat{\mathbf{z}}_n\}$ from a prior $P(\hat{\mathbf{z}})$.

   Update the network parameters of $G, \widetilde{D}, F$ by descending with their stochastic gradients:

$$\nabla_{\Theta_{G,\widetilde{D},F}} \left\{ \begin{array}{l} \frac{1}{n}\sum_{i=1}^{n} \beta[\log(1 - D(G(\hat{\mathbf{z}}_i)))] \quad + \\ \frac{1}{m}\sum_{i=1}^{m} \left[ \ell_c\big(F(\mathbf{x}_i), \text{ReLU}(\mathbf{y}_i - \widetilde{D}(\mathbf{x}_i))\big) + \ell_g\big(G(\text{ReLU}(\mathbf{y}_i - \widetilde{D}(\mathbf{x}_i))), \mathbf{x}_i\big) \right] \end{array} \right\}$$

   **for** r=1:$k$ **do**

      Sample $n$ label vectors $\{\hat{\mathbf{z}}_1, \cdots, \hat{\mathbf{z}}_n\}$ from a prior $P(\hat{\mathbf{z}})$.

      Update the parameters of the discrimination network by ascending with its stochastic gradient:

$$\nabla_{\Theta_D} \ \beta \left[ \frac{1}{m}\sum_{i=1}^{m}[\log D(\mathbf{x}_i)] + \frac{1}{n}\sum_{i=1}^{n}[\log(1 - D(G(\hat{\mathbf{z}})))] \right]$$

   **end for**
**end for**

---

**Proposition 1.** *For any $G$, $\widetilde{D}$, and $F$, the optimal discriminator $D$ is given by*

$$D^*_{G,\widetilde{D},F}(\mathbf{x}) = D^*_G(\mathbf{x}) = p_S(\mathbf{x})/\big(p_S(\mathbf{x}) + p_g(\mathbf{x})\big) \tag{5}$$

*where $p_S(\cdot)$ and $p_g(\cdot)$ denote the probability distributions of real and generated data respectively.*

*Proof.* Due to the conditional independence relationship between $D$ and $\{F, \widetilde{D}\}$, the optimal discriminator $D$ only depends on the generator $G$. Given fixed $G$, the optimal discriminator can be derived in the same way as in the standard GAN model (Goodfellow et al., 2014, Proposition 1). □

**Proposition 2.** *Assume the model has sufficient capacity. Let $C(G, \widetilde{D}, F) = \max_D \mathcal{L}(G, \widetilde{D}, F, D)$. Given fixed $\widetilde{D}$, the minimum of $C(G, \widetilde{D}, F)$ is lower bounded by $\mathbb{E}_{(\mathbf{x}_i, \mathbf{y}_i) \sim S} \, H\big(ReLU(\mathbf{y}_i - \widetilde{D}(\mathbf{x}_i))\big) - \beta \log 4$, which can be achieved when $F(\mathbf{x}_i) = ReLU(\mathbf{y}_i - \widetilde{D}(\mathbf{x}_i))$, $G(F(\mathbf{x}_i)) = \mathbf{x}_i$, and $p_g = p_S$.*

Here $H(\cdot)$ denotes an entropy function. Proof is provided in the appendix. This proposition suggests that $F$ and $G$ should be inverse mapping functions for each other in the ideal optimal case.

## 5 EXPERIMENTS

### 5.1 EXPERIMENTAL SETTING

**Datasets.** We conducted experiments on twelve multi-label classification datasets. Three of them have existing partial multi-label learning settings (mirflickr, music_style and music_emotion (Fang & Zhang, 2019)). For each of the other nine datasets (Zhang & Zhou, 2014), we transformed it into a PML dataset by randomly adding irrelevant labels into the candidate label set of each training instance. By adding different numbers of irrelevant labels, for each dataset we can create multiple PML variants with different average numbers of candidate labels. Following the setting of (Xie & Huang, 2018), we also filtered out the rare labels and kept at most 15 classes in each dataset. The detailed characteristics of the processed datasets are given in Table 1.

**Comparison Methods.** We compared our proposed method with four state-of-the-art PML methods and one baseline multi-label learning method. We adopted a simple but effective neural network based multi-label learning method, ML-RBF Zhang (2009), derived from the radial basis function (RBF), as a baseline method, which performs PML by treating all the candidate labels as ground-truth labels. Then we used four recently developed PML methods for comparison, including the PML-LC and PML-FP methods from (Xie & Huang, 2018) and the PARTICLE-VLS and PARTICLE-MAP methods from (Fang & Zhang, 2019).

**Implementation.** The proposed PML-GAN model has four component networks, all of which are designed as multilayer perceptrons with Leaky ReLu activation for the middle layers. The

Table 1: Information of the experimental data sets. The number of instances, features and classes are recorded. The "*avg.#CLs*" column lists the average number of candidate labels in each PML set.

| Dataset | #Inst. | #Feats | #Classes | avg.#CLs | Dataset | #Inst. | #Feats | #Classes | avg.#CLs |
|---------|--------|--------|----------|----------|---------|--------|--------|----------|----------|
| music_emotion | 6833 | 98 | 11 | 5.29 | music_style | 6839 | 98 | 10 | 6.04 |
| mirflickr | 10433 | 100 | 7 | 3.35 | image | 2000 | 294 | 5 | 2,3,4 |
| scene | 2407 | 294 | 6 | 3,4,5 | yeast | 2417 | 103 | 14 | 9,10,11,12 |
| enron | 1702 | 1001 | 15 | 8,9,10,11,12,13 | corel5k | 5000 | 499 | 15 | 8,9,10,11,12,13 |
| eurlex_dc | 8636 | 100 | 15 | 8,9,10,11,12,13 | eurlex_sm | 12679 | 100 | 15 | 8,9,10,11,12,13 |
| delicious | 14000 | 500 | 15 | 8,9,10,11,12,13 | tmc2007 | 28596 | 49060 | 15 | 8,9,10,11,12,13 |

disambiguator, predictor, and discriminator are all three-layer networks with sigmoid activation in the output layer, while the generator is a five layer network with Tanh activation in the output layer. Batch normalization is also deployed in the middle three layers of the generation network. We used the Adam (Kingma & Ba, 2014) optimizer in our implementation. The mini-batch size, $m$, is set to 64. The hyperparameters $k$ (the number of steps for discriminator update) and $n$ (the number of label vectors sampled) in Algorithm 1 are set to 1 and $2^{10}$ respectively. The hyperparameter $\beta$ is chosen from $\{0.001, 0.01, 0.1, 1, 10\}$ based on the classification loss value $\mathcal{L}_c$ in the training objective function; that is, the $\beta$ value that leads to the smallest training $\mathcal{L}_c$ loss will be chosen.

## 5.2 COMPARISON RESULTS

We compared the proposed PML-GAN method with the five comparison methods on the twelve datasets. For each dataset, we randomly select 80% of the data for training and use the remaining 20% for testing. We repeat each experiment 10 times with different random partitions of the datasets. The comparison test results in terms of four commonly used evaluation metrics (Hamming loss, ranking loss, average precision, and macro-averaging AUC) (Zhang & Zhou, 2014) are reported in Table 2. The results are the means and standard deviations over the 10 repeated runs. We can see that the methods specially developed for PML problems all outperform the baseline multi-label neural network classifier, ML-RBF, in most cases. But it is difficult to beat the baseline competitor on all the datasets with different evaluation metrics. Among the total 48 cases over 12 datasets and 4 evaluation metrics, PARTICLE-VLS, PARTICLE-MAP, PML-LC and PML-FP outperform ML-RBF in 44, 48, 37 and 45 cases respectively. By contrast, the proposed PML-GAN method outperforms ML-RBF *consistently* across all the 48 cases with remarkably performance gains. Even comparing with all the other four PML methods, PML-GAN produced the best results in 43 out of the total 48 cases. Moreover, the performance gains yield by PML-GAN over all the other methods are quite notable in many cases. For example, in terms of average precision, PML-GAN outperforms the best alternative comparison method by 4.6%, 4.1%, and 3.3% on *eurlex_dc, scene* and *image* respectively. These results clearly demonstrate the effectiveness of the proposed PML-GAN model.

The results reported above are produced on each dataset with a selected average number of candidate labels. As shown in Table 1, we have multiple PML variants with different numbers of candidate labels for each of the nine datasets in the list (except music_emotion, music_style, and mirflickr). In total this provides us 49 PML datasets. We hence also conducted experiments on each of these 49 variant datasets, by comparing the proposed PML-GAN with each of the other methods in terms of the 4 evaluation metrics. In total there are 196 comparison cases for each pair of methods. For the comparison of "PML-GAN vs other method" in each case, we conducted pairwise t-test at significance level of 0.05. The win/tie/loss counts in all cases are reported in Table 3. We can see that overall the proposed PML-GAN significantly outperforms PARTICLE-VLS, PARTICLE-MAP, PML-LC, PML-FP, and ML-RBF in 77%, 76%, 78.5%, 81.6%, and 90.3% of the cases respectively. This again validates the efficacy of the proposed method.

**Impact of Irrelevant Labels**    To demonstrate how would the number of irrelevant labels affect the performance of PML methods, we plotted the experimental results on the *delicious* dataset with different average numbers of candidate labels in Figure 4. We can see with the increase of the number of irrelevant labels, consequently the average number of candidate labels, the performance of each method in general degrades. Nevertheless, the proposed PML-GAN consistently outperforms all the other methods. Moreover, in terms of Hamming loss, the performance of PML-GAN actually is quite stable with the increase of the noisy labels. This validates the effectiveness of PML-GAN in irrelevant noisy label disambiguation.

Table 2: Comparison results of in terms of Hamming loss, ranking loss, average precision, and micro-averaging AUC. The best results are presented in bold font. The average number of candidate labels is presented under the column "avg.#C.Ls".

| Data set | avg.#C.Ls | PML-GAN | PARTICLE-VLS | PARTICLE-MAP | PML-LC | PML-FP | ML-RBF |
|---|---|---|---|---|---|---|---|
| Hamming loss (the smaller, the better) | | | | | | | |
| music_emotion | 5.29 | **.197±.004** | .212±.004 | .215±.004 | .236±.003 | .245±.004 | .779±.004 |
| music_style | 6.04 | **.116±.005** | .121±.003 | .175±.005 | .126±.004 | .126±.004 | .856±.001 |
| mirflickr | 3.35 | **.171±.002** | .178±.035 | .189±.081 | .202±.057 | .202±.057 | .748±.002 |
| image | 3 | **.202±.008** | .234±.065 | .269±.096 | .264±.072 | .267±.063 | .754±.003 |
| scene | 4 | **.137±.007** | .184±.037 | .174±.035 | .178±.029 | .187±.038 | .820±.001 |
| yeast | | **.216±.005** | .226±.004 | .220±.008 | .226±.008 | .219±.009 | .694±.003 |
| enron | | **.186±.004** | .197±.032 | .190±.036 | .206±.027 | .206±.027 | .813±.004 |
| corel5k | | **.119±.001** | .189±.012 | .269±.027 | .151±.008 | .152±.008 | .886±.001 |
| eurlex_dc | 10 | **.044±.001** | .061±.001 | .064±.004 | .096±.001 | .071±.001 | .933±.001 |
| eurlex_sm | | .082±.001 | **.067±.001** | .076±.002 | .119±.006 | .122±.002 | .885±.001 |
| delicious | | **.248±.003** | .260±.003 | .290±.005 | .290±.004 | .290±.004 | .712±.002 |
| tmc2007 | | **.085±.001** | .090±.003 | .110±.003 | .103±.002 | .103±.002 | .857±.001 |
| Ranking loss (the smaller, the better) | | | | | | | |
| music_emotion | 5.29 | .243±.009 | .263±.008 | **.240±.007** | .267±.009 | .275±.010 | .365±.010 |
| music_style | 6.04 | **.141±.001** | .163±.007 | .147±.005 | .215±.005 | .150±.005 | .242±.006 |
| mirflickr | 3.35 | **.127±.014** | .227±.029 | .129±.108 | .160±.029 | .143±.028 | .195±.015 |
| image | 3 | **.192±.015** | .239±.077 | .250±.085 | .291±.134 | .217±.120 | .251±.019 |
| scene | 4 | **.131±.014** | .177±.049 | .167±.060 | .192±.032 | .238±.056 | .188±.014 |
| yeast | | **.193±.008** | .203±.007 | .208±.012 | .219±.011 | .203±.008 | .270±.007 |
| enron | | **.179±.013** | .240±.078 | .182±.029 | .239±.048 | .239±.047 | .244±.010 |
| corel5k | | **.293±.012** | .367±.032 | .311±.008 | .366±.035 | .398±.025 | .404±.082 |
| eurlex_dc | 10 | **.065±.003** | .150±.004 | .085±.004 | .137±.008 | .131±.001 | .135±.003 |
| eurlex_sm | | **.119±.005** | .129±.007 | .127±.009 | .282±.007 | .182±.008 | .183±.003 |
| delicious | | **.256±.006** | .314±.005 | .276±.004 | .277±.005 | .276±.005 | .316±.003 |
| tmc2007 | | **.071±.003** | .096±.008 | .095±.007 | .082±.005 | .080±.005 | .153±.002 |
| Average precision (the larger, the better) | | | | | | | |
| music_emotion | 5.29 | **.621±.012** | .605±.012 | .612±.009 | .574±.013 | .568±.014 | .506±.012 |
| music_style | 6.04 | **.734±.015** | .715±.009 | .709±.009 | .702±.008 | .703±.008 | .646±.010 |
| mirflickr | 3.35 | .771±.026 | .678±.027 | **.791±.202** | .736±.043 | .758±.039 | .676±.048 |
| image | 3 | **.774±.013** | .741±.090 | .729±.086 | .644±.131 | .725±.119 | .723±.021 |
| scene | 4 | **.794±.014** | .750±.074 | .753±.064 | .689±.047 | .710±.079 | .728±.015 |
| yeast | | **.733±.008** | .724±.010 | .714±.010 | .721±.012 | .728±.010 | .634±.008 |
| enron | | **.665±.020** | .595±.099 | .661±.047 | .556±.041 | .575±.041 | .560±.009 |
| corel5k | | **.440±.014** | .377±.025 | .415±.008 | .345±.027 | .384±.021 | .334±.008 |
| eurlex_dc | 10 | **.797±.009** | .692±.013 | .751±.008 | .693±.019 | .716±.014 | .710±.000 |
| eurlex_sm | | **.722±.002** | .705±.009 | .683±.011 | .438±.016 | .679±.011 | .656±.000 |
| delicious | | **.630±.007** | .596±.007 | .601±.008 | .607±.007 | .608±.006 | .576±.004 |
| tmc2007 | | **.820±.004** | .799±.013 | .759±.013 | .793±.012 | .794±.012 | .662±.003 |
| Macro-averaging AUC (the larger, the better) | | | | | | | |
| music_emotion | 5.29 | **.726±.006** | .673±.008 | .676±.004 | .632±.008 | .636±.007 | .504±.003 |
| music_style | 6.04 | .706±.016 | .694±.007 | .715±.005 | .754±.006 | **.755±.006** | .503±.003 |
| mirflickr | 3.35 | **.873±.002** | .806±.040 | .816±.057 | .842±.030 | .840±.031 | .808±.000 |
| image | 3 | **.798±.015** | .735±.119 | .757±.103 | .727±.131 | .728±.130 | .729±.000 |
| scene | 4 | **.874±.012** | .789±.054 | .814±.049 | .734±.041 | .735±.038 | .730±.010 |
| yeast | | **.666±.026** | .610±.017 | .650±.014 | .631±.019 | .636±.020 | .600±.012 |
| enron | | **.668±.018** | .607±.024 | .660±.031 | .631±.029 | .631±.029 | .596±.016 |
| corel5k | | **.628±.007** | .542±.039 | .618±.031 | .550±.033 | .551±.034 | .527±.007 |
| eurlex_dc | 10 | **.872±.009** | .867±.011 | .835±.008 | .840±.005 | .853±.007 | .825±.006 |
| eurlex_sm | | .827±.008 | **.843±.008** | .823±.008 | .799±.007 | .780±.004 | .774±.003 |
| delicious | | **.712±.004** | .666±.006 | .634±.006 | .688±.006 | .688±.006 | .632±.002 |
| tmc2007 | | **.886±.002** | .836±.010 | .830±.009 | .861±.008 | .859±.008 | .765±.003 |

## 5.3 ABLATION STUDY

As shown in Eq.(6), the objective of PML-GAN contains three parts: classification loss, generation loss and adversarial loss. The generation loss and adversarial loss are integrated to assist the predictor training. To investigate and validate the contribution of the generation loss and adversarial loss, we conducted an ablation study by comparing PML-GAN with three of its ablation variants: (1) CLS-GEN, which drops the adversarial loss; (2) CLS-GAN, which drops the generation loss; and (3) CLS-ML, which only uses the classification loss by dropping both the adversarial loss and generation loss. The comparison results are reported in Table 4. We can see that comparing to the full model, all three variants produced inferior results in general. Among the three variants, both *CLS-GEN* and *CLS-GAN* outperform *CLS-ML* in most cases. This suggests that both generation loss and adversarial

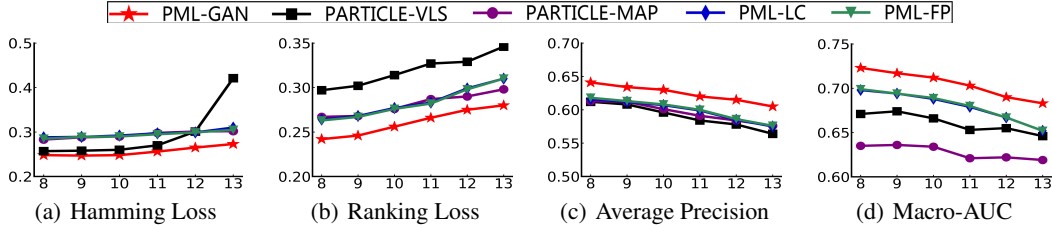

|  | (a) Hamming Loss | (b) Ranking Loss | (c) Average Precision | (d) Macro-AUC |
|---|---|---|---|---|

Figure 4: Test results with different average numbers of candidate labels on the delicious dataset.

Table 3: Win/tie/loss counts of pairwise t-test (with $p < 0.05$ ) between PML-GAN and each comparison method over all dataset variants with different numbers of candidate labels.

| Evaluation Metric | PML-GAN vs − | | | | |
|---|---|---|---|---|---|
| | PARTICLE-VLS | PARTICLE-MAP | PML-LC | PML-FP | ML-RBF |
| Hamming loss | 39/6/4 | 38/9/2 | 40/7/2 | 40/3/6 | 45/4/0 |
| Ranking loss | 38/9/2 | 38/8/3 | 38/8/3 | 40/5/4 | 44/3/2 |
| Average precision | 37/8/4 | 36/10/3 | 40/6/3 | 42/4/3 | 46/3/0 |
| Macro-averaging AUC | 37/7/5 | 37/8/4 | 36/7/6 | 38/5/6 | 42/7/0 |
| Total | 151/30/15 | 149/35/12 | 154/28/14 | 160/17/19 | 177/17/2 |

Table 4: Comparison results of PML-GAN and its three ablation variants.

| Data set | PML-GAN | CLS-GEN | CLS-GAN | CLS-ML | PML-GAN | CLS-GEN | CLS-GAN | CLS-ML |
|---|---|---|---|---|---|---|---|---|
| Hamming loss (the smaller, the better) | | | | | Ranking loss (the smaller, the better) | | | |
| music_emotion | **.197**±.004 | .200±.003 | .199±.005 | .202±.003 | **.243**±.009 | .247±.008 | .244±.007 | .248±.007 |
| music_style | **.116**±.005 | .117±.003 | .118±.003 | .120±.003 | **.141**±.001 | .146±.004 | .143±.007 | .164±.007 |
| mirflickr | **.171**±.002 | .174±.005 | .172±.005 | .176±.004 | **.127**±.014 | .132±.022 | .128±.020 | .135±.015 |
| image | **.202**±.008 | .205±.006 | .208±.008 | .230±.008 | **.192**±.015 | .196±.013 | .196±.013 | .206±.014 |
| scene | **.137**±.007 | .140±.013 | .143±.009 | .152±.007 | **.131**±.014 | .136±.009 | .131±.007 | .137±.006 |
| yeast | **.216**±.005 | .219±.004 | .217±.006 | .228±.005 | **.193**±.008 | .198±.010 | .196±.010 | .205±.012 |
| enron | **.186**±.004 | .281±.014 | .273±.018 | .281±.012 | **.179**±.013 | .184±.013 | .184±.008 | .189±.010 |
| corel5k | **.119**±.001 | **.119**±.002 | .121±.002 | .122±.002 | **.293**±.012 | .302±.009 | .302±.017 | .308±.009 |
| eurlex_dc | **.044**±.001 | .052±.003 | .046±.001 | .054±.001 | **.065**±.003 | .085±.012 | .068±.003 | .071±.005 |
| eurlex_sm | **.082**±.001 | .083±.001 | .084±.001 | .086±.001 | **.119**±.005 | .120±.003 | .123±.005 | .125±.004 |
| delicious | **.248**±.003 | .251±.001 | .249±.002 | .255±.002 | **.256**±.006 | .259±.006 | .257±.004 | .323±.006 |
| tmc2007 | **.085**±.001 | .088±.001 | **.085**±.001 | .091±.001 | **.071**±.003 | .074±.002 | .072±.002 | .075±.003 |
| Average precision (the larger, the better) | | | | | Macro-averaging AUC (the larger, the better) | | | |
| music_emotion | **.621**±.012 | .608±.012 | .620±.010 | .614±.010 | **.726**±.006 | .724±.007 | .724±.009 | .615±.010 |
| music_style | **.734**±.015 | .726±.008 | .730±.010 | .700±.012 | **.706**±.016 | .700±.011 | .703±.000 | .691±.014 |
| mirflickr | **.771**±.026 | .763±.042 | .769±.038 | .730±.010 | **.873**±.002 | .872±.003 | .872±.002 | .860±.003 |
| image | **.774**±.013 | .771±.015 | .766±.017 | .758±.018 | **.798**±.015 | .797±.009 | .794±.012 | .790±.000 |
| scene | **.794**±.014 | **.794**±.013 | .789±.011 | .783±.009 | **.874**±.012 | .872±.006 | .873±.009 | .852±.010 |
| yeast | **.733**±.008 | .728±.014 | .731±.015 | .708±.012 | **.666**±.026 | .628±.015 | .627±.011 | .625±.011 |
| enron | **.665**±.020 | .663±.025 | .657±.014 | .655±.023 | **.668**±.018 | .566±.012 | .573±.013 | .561±.008 |
| corel5k | **.440**±.014 | .434±.011 | .436±.015 | .432±.008 | **.628**±.007 | .579±.004 | .579±.006 | .570±.006 |
| eurlex_dc | **.797**±.009 | .781±.019 | **.797**±.008 | .779±.011 | **.872**±.009 | .839±.016 | **.872**±.009 | .822±.007 |
| eurlex_sm | **.722**±.002 | .719±.003 | .714±.009 | .712±.004 | **.827**±.008 | .824±.005 | .825±.008 | .820±.005 |
| delicious | **.630**±.007 | .628±.004 | **.630**±.006 | .628±.005 | **.712**±.004 | .710±.003 | .711±.002 | .709±.002 |
| tmc2007 | **.820**±.004 | .818±.004 | **.820**±.004 | .815±.003 | **.886**±.002 | .884±.003 | **.886**±.002 | .883±.002 |

loss are critical terms for the proposed model. Moreover, even the baseline variant *CLS-ML* still produces some reasonable PML results. This suggests the integration of our proposed prediction network and disambiguation network is also effective.

# 6    CONCLUSION

In this paper, we proposed a novel adversarial model for partial multi-label learning. The proposed model comprises four component networks, which form an encoder-decoder framework to improve noisy label disambiguation and boost multi-label learning performance. The training problem forms a minimax adversarial optimization, which is solved using an alternative min-max procedure with minibatch-based stochastic gradient descent. We conducted extensive experiments on multiple PML datasets. The results show that the proposed model outperforms all the comparison methods and achieves the state-of-the-art PML performance.

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

# A APPENDIX

In this appendix, we provide proof for Proposition 2, details of the network structure for the PML-GAN model, and additional experimental results. For additional experimental results, we first reported results with two additional evaluation metrics, and results of parameter sensitivity analysis. Then we empirically compared PML-GAN with its variant that considers additional classification loss on the generated data. Finally we conducted extended experiments on a few datasets where we constructed PML data by adding large number of label noise.

## A.1 PROOF OF PROPOSITION 2

*Proof.* As shown in section 3.3, $\mathcal{L}(G, \widetilde{D}, F, D)$ denotes the objective function of PML-GAN. Based on the solution for optimal discriminator $D$ in Proposition 1, we have:

$$
\begin{aligned}
C(G, \widetilde{D}, F) &= \max_D \mathcal{L}(G, \widetilde{D}, F, D) \\
&= \left\{ \begin{aligned} &\mathbb{E}_{(\mathbf{x}_i, \mathbf{y}_i) \sim S}\left(\ell_c(F(\mathbf{x}_i), \mathrm{ReLU}(\mathbf{y}_i - \widetilde{D}(\mathbf{x}_i))) + \ell_g(G(\mathrm{ReLU}(\mathbf{y}_i - \widetilde{D}(\mathbf{x}_i))), \mathbf{x}_i)\right) + \\ &\beta\left(\mathbb{E}_{\mathbf{x} \sim p_S}[\log D_G^*(\mathbf{x})] + \mathbb{E}_{\hat{\mathbf{z}} \sim P(\hat{\mathbf{z}})}[\log(1 - D_G^*(G(\hat{\mathbf{z}})))]\right) \end{aligned} \right\} \\
&= \left\{ \begin{aligned} &\mathbb{E}_{(\mathbf{x}_i, \mathbf{y}_i) \sim S}\left(\ell_c(F(\mathbf{x}_i), \mathrm{ReLU}(\mathbf{y}_i - \widetilde{D}(\mathbf{x}_i))) + \ell_g(G(\mathrm{ReLU}(\mathbf{y}_i - \widetilde{D}(\mathbf{x}_i))), \mathbf{x}_i)\right) + \\ &\beta\left(\mathbb{E}_{\mathbf{x} \sim p_S}[\log D_G^*(\mathbf{x})] + \mathbb{E}_{\mathbf{x} \sim p_g}[\log(1 - D_G^*(\mathbf{x}))]\right) \end{aligned} \right\} \\
&= \left\{ \begin{aligned} &\mathbb{E}_{(\mathbf{x}_i, \mathbf{y}_i) \sim S}\left(\ell_c(F(\mathbf{x}_i), \mathrm{ReLU}(\mathbf{y}_i - \widetilde{D}(\mathbf{x}_i))) + \ell_g(G(\mathrm{ReLU}(\mathbf{y}_i - \widetilde{D}(\mathbf{x}_i))), \mathbf{x}_i)\right) + \\ &\beta\left(\mathbb{E}_{\mathbf{x} \sim p_S}[\log \frac{p_S(\mathbf{x})}{p_S(\mathbf{x}) + p_g(\mathbf{x})}] + \mathbb{E}_{\mathbf{x} \sim p_g}[\log \frac{p_g(\mathbf{x})}{p_S(\mathbf{x}) + p_g(\mathbf{x})}]\right) \end{aligned} \right\}
\end{aligned}
$$

Note given fixed $\widetilde{D}$, $F$ is conditionally independent from $G$ and $D$. Hence the minimization of $C(G, \widetilde{D}, F)$ over $F$ can be independently conducted from the minimization over $G$.

Let $\mathbf{z}_i = \mathrm{ReLU}(\mathbf{y}_i - \widetilde{D}(\mathbf{x}_i))$. With the cross-entropy loss function $\ell_c(\cdot, \cdot)$ we have:

$$
\begin{aligned}
\min_F C(G, \widetilde{D}, F) &\equiv \min_F \mathbb{E}_{(\mathbf{x}_i, \mathbf{y}_i) \sim S} \ell_c\left(F(\mathbf{x}_i), \mathrm{ReLU}(\mathbf{y}_i - \widetilde{D}(\mathbf{x}_i))\right) \\
&\equiv \min_F \mathbb{E}_{(\mathbf{x}_i, \mathbf{y}_i) \sim S}\left[-\mathbf{z}_i^\top \log F(\mathbf{x}_i) - (1 - \mathbf{z}_i)^\top \log(1 - F(\mathbf{x}_i))\right] \\
&\equiv \min_F \mathbb{E}_{(\mathbf{x}_i, \mathbf{y}_i) \sim S} H(\mathbf{z}_i) + \mathrm{KL}(\mathbf{z}_i \parallel F(\mathbf{x}_i)) \\
&\geq \mathbb{E}_{(\mathbf{x}_i, \mathbf{y}_i) \sim S} H(\mathbf{z}_i)
\end{aligned}
$$

where $H(\cdot)$ denotes the entropy over a binomial distriubtion vector and $\mathrm{KL}(\cdot)$ denotes the KL-divergence between two sets of binomial distributions. Assume sufficient capacity for $F$, the minimum can be reached when the predictor obtains the same distributions as the $\mathbf{z}_i$; that is

$$
F^*(\mathbf{x}_i) = \mathbf{z}_i = \mathrm{ReLU}(\mathbf{y}_i - \widetilde{D}(\mathbf{x}_i)), \quad \forall(\mathbf{x}_i, \mathbf{y}_i) \in S
$$

Next let's consider the minimization problem over $G$. Note $G$ is involved in both the generation loss and adversarial loss. If we could find solutions that lead to minimals in both losses separately, we can guarantee a minimal in the united loss. The adversarial loss part in $C(G, \widetilde{D}, F)$ can be rewritten as

$$
\begin{aligned}
\beta \mathcal{L}_{adv} &= \beta\left(\mathbb{E}_{\mathbf{x} \sim p_S}[\log \frac{p_S(\mathbf{x})}{p_S(\mathbf{x}) + p_g(\mathbf{x})}] + \mathbb{E}_{\mathbf{x} \sim p_g}[\log \frac{p_g(\mathbf{x})}{p_S(\mathbf{x}) + p_g(\mathbf{x})}]\right) \\
&= \beta\left(\mathrm{KL}(p_S, \frac{p_S + p_g}{2}) - \log 2 + \mathrm{KL}(p_g, \frac{p_S + p_g}{2}) - \log 2\right) \\
&= \beta\left(\mathrm{KL}(p_S, \frac{p_S + p_g}{2}) + \mathrm{KL}(p_g, \frac{p_S + p_g}{2}) - \log 4\right) \\
&\geq -\beta \log 4
\end{aligned}
$$

Table 5: The network architecture of PML-GAN. BN: Batch normalization; LReLU: Leaky rectified unit; Act.: Activation function; dim: Feature dimension of training samples **x**; class num: the number of class labels.

| Generator $G$ | | | | Discriminator $D$ | | |
|---|---|---|---|---|---|---|
| Input | Output | BN | Act. | Input | Output | Act. |
| $\hat{z}$ | 512 | $\times$ | LReLU | data | 512 | LReLU |
| 512 | 1024 | $\checkmark$ | LReLU | 512 | 256 | LReLU |
| 1024 | 256 | $\checkmark$ | LReLU | 256 | 1 | Sigmoid |
| 256 | 128 | $\checkmark$ | LReLU | | | |
| 128 | dim | $\times$ | Tanh | | | |
| **Disambiguator** $\widetilde{D}$ | | | | **Predictor** $F$ | | |
| $x$ | 512 | – | LReLU | $x$ | 512 | LReLU |
| 512 | 256 | – | LReLU | 512 | 256 | LReLU |
| 256 | class num | – | Sigmoid | 256 | class num | Sigmoid |

where the minimal can be achieved when $p_S = p_g$ which leads to zeros as the KL-divergence values. The generation loss part (with least squares loss function) in $C(G, \widetilde{D}, F)$ can be rewritten as

$$\mathbb{E}_{(\mathbf{x}_i, \mathbf{y}_i) \sim S} \Big( \ell_g(G(\text{ReLU}(\mathbf{y}_i - \widetilde{D}(\mathbf{x}_i))), \mathbf{x}_i) \Big)$$

$$= \mathbb{E}_{(\mathbf{x}_i, \mathbf{y}_i) \sim S} \| G(\text{ReLU}(\mathbf{y}_i - \widetilde{D}(\mathbf{x}_i))) - \mathbf{x}_i \|^2$$

$$\geq 0$$

where the minimal 0 can only be achieved when

$$G(\text{ReLU}(\mathbf{y}_i - \widetilde{D}(\mathbf{x}_i))) = \mathbf{x}_i, \quad \forall (\mathbf{x}_i, \mathbf{y}_i) \in S$$

It is obvious the optimal condition above can be satisfied simultaneously together with the condition $p_g = p_S$. Hence the proposition is proved. $\square$

## A.2 NETWORK STRUCTURE OF THE PML-GAN MODEL

The proposed PML-GAN model has four component networks, all of them are designed as multilayer perceptrons with LeakyReLu activation function for the middle layers. The disambiguator, predictor, and discriminator are all three-layer networks with sigmoid activation in the output layer, while the generator is a five layer network with Tanh activation in the output layer. Batch normalization is also deployed in the middle three layers of the generation network. The detailed input and output setup information for each layer of the networks is given in Table 5.

## A.3 ADDITIONAL EXPERIMENTAL RESULTS IN MORE EVALUATION METRICS

For the comparison results, we evaluated the test performance using six commonly used metrics from Zhang & Zhou (2014). Due to space limitation, we previously reported the results in terms of four metrics. The results in terms of the remaining two metrics, one error and coverage, are reported in Table 6. We can see that our proposed PML-GAN again produced the best results in majority cases.

In Table 7, we reported the ablation study results in terms of coverage and one error. The relative comparison performance is quite similar to the results reported in the paper. It suggests the component networks in our model are necessary and useful.

## A.4 PARAMETER SENSITIVITY ANALYSIS

We also conducted experiments on the eurlex_dc data set to study the impact of the trade-off hyperparameter $\beta$ on the performance of PML-GAN. We repeated the experiments in the same setting as before with different $\beta$ values from the range $\{0.001, 0.01, 0.1, 1, 10\}$. Note a larger $\beta$ provides larger weight to the adversarial loss.

Figure 5 reports the average test results over 10 runs for different $\beta$ values. We can see when $\beta$ is very small, the performance is poor. With the increase of $\beta$, the performance improves. This suggests the adversarial loss is important. When $\beta = 0.1$ the best result is obtained. When $\beta$ is too large, performance degrades as the adversarial loss dominates. This makes sense since the adversarial loss is expected to help the classification model, rather than dominating the learning.

Table 6: Comparison results for PML-GAN and the other methods in terms of two additional metrics: coverage and one error. The best results are in bold font.

| Data set | avg.#C.Ls | PML-GAN | PARTICLE-VLS | PARTICLE-MAP | PML-LC | PML-FP | ML-RBF |
|---|---|---|---|---|---|---|---|
| Coverage (the smaller, the better) | | | | | | | |
| music_emotion | 5.29 | .407±.008 | .413±.007 | **.396±.007** | .428±.010 | .434±.010 | .508±.006 |
| music_style | 6.04 | **.202±.011** | .208±.006 | .205±.007 | .203±.008 | .203±.008 | .305±.006 |
| mirflickr | 3.35 | **.227±.010** | .270±.079 | **.227±.057** | .234±.073 | .233±.074 | .258±.002 |
| image | 3 | **.211±.014** | .227±.079 | .263±.085 | .305±.109 | .298±.095 | .249±.015 |
| scene | 4 | **.124±.010** | .144±.054 | .190±.050 | .180±.033 | .214±.045 | .167±.012 |
| yeast | | .502±.014 | .506±.010 | .528±.003 | .524±.016 | **.495±.016** | .583±.007 |
| enron | | .402±.015 | .457±.116 | **.369±.075** | .496±.089 | .496±.088 | .458±.017 |
| corel5k | | **.384±.011** | .541±.022 | .393±.017 | .497±.042 | .489±.035 | .492±.009 |
| eurlex_dc | 10 | **.063±.003** | .136±.010 | .082±.004 | .318±.017 | .162±.015 | .117±.003 |
| eurlex_sm | | **.172±.006** | .192±.009 | .183±.012 | .403±.009 | .408±.009 | .243±.005 |
| delicious | | **.564±.008** | .603±.004 | .584±.004 | .591±.005 | .590±.005 | .610±.004 |
| tmc2007 | | **.181±.006** | .205±.008 | .214±.011 | .196±.007 | .194±.008 | .283±.003 |
| One error (the smaller, the better) | | | | | | | |
| music_emotion | 5.29 | **.444±.027** | .473±.016 | .475±.018 | .556±.028 | .540±.027 | .587±.019 |
| music_style | 6.04 | **.346±.022** | .374±.005 | .399±.019 | .409±.013 | .408±.013 | .385±.006 |
| mirflickr | 3.35 | .337±.059 | **.165±.150** | .229±.306 | .300±.129 | .298±.121 | .338±.002 |
| image | 3 | **.340±.020** | .369±.134 | .387±.147 | .542±.191 | .549±.174 | .398±.034 |
| scene | 4 | **.329±.020** | .340±.078 | .349±.082 | .497±.089 | .523±.118 | .428±.022 |
| yeast | | .250±.019 | **.248±.019** | .252±.018 | .257±.017 | .263±.027 | .408±.023 |
| enron | | **.307±.039** | .411±.101 | .351±.040 | .494±.039 | .498±.038 | .495±.019 |
| corel5k | | **.697±.020** | .835±.025 | .721±.035 | .784±.029 | .787±.024 | .809±.015 |
| eurlex_dc | 10 | **.305±.016** | .390±.016 | .374±.014 | .707±.014 | .518±.011 | .342±.008 |
| eurlex_sm | | **.341±.003** | .350±.014 | .360±.015 | .506±.031 | .542±.018 | .340±.005 |
| delicious | | .372±.012 | **.366±.015** | .414±.018 | .401±.015 | .399±.013 | .450±.009 |
| tmc2007 | | .200±.005 | **.194±.029** | .267±.018 | .235±.019 | .236±.019 | .388±.006 |

Table 7: Comparison of PML-GAN with its ablation variants in terms of coverage and one error. The best results are in bold font.

| Data set | PML-GAN | CLS-GEN | CLS-GAN | CLS-ML | PML-GAN | CLS-GEN | CLS-GAN | CLS-ML |
|---|---|---|---|---|---|---|---|---|
| Coverage (the smaller, the better) | | | | | One error (the smaller, the better) | | | |
| music_emotion | **.407±.008** | .410±.008 | .408±.006 | .410±.005 | **.444±.027** | .460±.022 | .446±.000 | .463±.024 |
| music_style | **.202±.011** | .204±.004 | .204±.008 | .207±.008 | **.346±.022** | .359±.014 | .351±.017 | .361±.017 |
| mirflickr | **.227±.010** | .230±.017 | **.227±.014** | .242±.013 | **.337±.059** | .388±.092 | .370±.085 | .405±.094 |
| image | **.211±.014** | .214±.011 | .212±.011 | .220±.011 | **.340±.020** | .344±.023 | .360±.032 | .368±.024 |
| scene | **.124±.010** | .127±.008 | .125±.006 | .128±.010 | **.329±.020** | .331±.023 | .341±.022 | .348±.018 |
| yeast | **.502±.014** | .509±.016 | .507±.013 | .509±.023 | **.250±.019** | .260±.027 | .251±.029 | .254±.024 |
| enron | **.402±.015** | .410±.019 | .408±.016 | .414±.015 | **.307±.039** | .309±.029 | .321±.027 | .333±.028 |
| corel5k | **.384±.011** | .394±.008 | .396±.017 | .399±.011 | **.697±.020** | .699±.017 | .698±.020 | .700±.016 |
| eurlex_dc | **.063±.003** | .082±.011 | **.063±.003** | .085±.005 | **.305±.016** | .312±.024 | .307±.013 | .312±.017 |
| eurlex_sm | **.172±.006** | .174±.004 | .176±.005 | .178±.004 | **.341±.003** | .343±.007 | .350±.015 | .351±.007 |
| delicious | **.564±.008** | .566±.008 | **.564±.006** | .628±.005 | **.372±.012** | .373±.010 | .373±.008 | .375±.011 |
| tmc2007 | **.181±.006** | .184±.004 | **.181±.004** | .185±.004 | **.200±.005** | .201±.005 | .203±.007 | .205±.004 |

## A.5 VARIANT OF PML-GAN WITH CLASSIFICATION LOSS ON GENERATED DATA

The classification loss in the PML-GAN model of Eq.(6) takes the real data into account, as that is the primary concern of PML. A natural extension is to consider an auxiliary classification loss on the generated data as well. This extension leads to the following variant of PML-GAN, which we denote as PML-GAN′:

$$
\min_{G,\widetilde{D},F} \max_{D} \quad \mathbb{E}_{(\mathbf{x}_i,\mathbf{y}_i)\sim S}\Big(\ell_c(F(\mathbf{x}_i),\mathbf{z}_i) + \ell_g(G(\mathbf{z}_i),\mathbf{x}_i)\Big) + \alpha\,\mathbb{E}_{\hat{\mathbf{z}}\sim P(\hat{\mathbf{z}})}\ell_c(F(G(\hat{\mathbf{z}})),\hat{\mathbf{z}}) +
$$

$$
\beta\Big(\mathbb{E}_{\mathbf{x}_i\sim S}[\log D(\mathbf{x}_i)] + \mathbb{E}_{\hat{\mathbf{z}}\sim P(\hat{\mathbf{z}})}[\log(1 - D(G(\hat{\mathbf{z}})))]\Big)  \tag{6}
$$

$$
\text{s.t.} \quad \mathbf{z}_i = \text{ReLU}(\mathbf{y}_i - \widetilde{D}(\mathbf{x}_i)), \quad \forall(\mathbf{x}_i,\mathbf{y}_i)\sim S
$$

where $\alpha$ and $\beta$ are trade-off hyperparameters. For the parameter selection of PML-GAN′, both $\alpha$ and $\beta$ are chosen from {0.001, 0.01, 0.1, 1, 10} based on the classification loss value $\mathcal{L}_c$ on the real data in the training objective function.

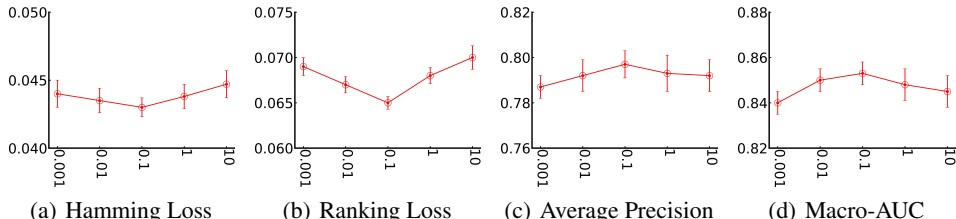

| (a) Hamming Loss | (b) Ranking Loss | (c) Average Precision | (d) Macro-AUC |

Figure 5: Test performance with different $\beta$ values on the eurlex_dc data set.

The comparison results between PML-GAN and PML-GAN′ are reported in Table 8. We can see that PML-GAN′ has similar performance with PML-GAN, and the difference between their results are very small in most cases. To statistically compare PML-GAN with PML-GAN′, we conducted pairwise t-test at significance level of 0.05 over all the 49 PML variant datasets in terms of the 4 evaluation metrics. The win/tie/loss counts in all 196 cases are reported in Table 9. We can see that overall the PML-GAN′ ties with PML-GAN in 99.5% cases. This suggests that an addition auxiliary classification loss on the generated data cannot improve PML-GAN, which makes sense given the goal is to learn a good classification model on the sufficient real data. Meanwhile, the additional hyperparameter $\alpha$ of PML-GAN′ induces much higher computational cost due to additional parameter selection. Overall, PML-GAN is a more suitable choice than PML-GAN′.

Table 8: Comparison results for PML-GAN and PML-GAN′ in terms of six evaluation metrics. The best results are presented in bold font. The average number of candidate labels is presented under column "avg.#C.Ls".

| Data set | avg.#C.Ls | PML-GAN | PML-GAN′ | PML-GAN | PML-GAN′ |
|---|---|---|---|---|---|
| | | Hamming loss (the smaller, the better) | | Ranking loss (the smaller, the better) | |
| music_emotion | 5.29 | **.197±.004** | .200±.003 | .243±.009 | **.239±.009** |
| music_style | 6.04 | .116±.005 | **.115±.002** | **.141±.001** | .145±.006 |
| mirflickr | 3.35 | **.171±.002** | **.171±.004** | .127±.014 | **.122±.025** |
| image | 3 | .202±.008 | **.200±.008** | .192±.015 | **.191±.010** |
| scene | 4 | .137±.007 | **.133±.007** | .131±.014 | **.119±.008** |
| yeast | | **.216±.005** | **.216±.005** | **.193±.008** | .194±.007 |
| enron | | **.186±.004** | **.186±.004** | .179±.013 | **.178±.012** |
| corel5k | | .119±.001 | **.113±.001** | **.293±.012** | .298±.014 |
| eurlex_dc | 10 | **.044±.001** | **.044±.001** | .065±.003 | **.064±.006** |
| eurlex_sm | | **.082±.001** | **.082±.001** | **.119±.005** | .119±.007 |
| delicious | | **.248±.003** | **.248±.003** | **.256±.006** | .257±.005 |
| tmc2007 | | **.085±.001** | .086±.001 | .071±.003 | **.069±.003** |
| | | Average precision (the larger, the better) | | Macro-averaging AUC (the larger, the better) | |
| music_emotion | 5.29 | **.621±.012** | **.621±.014** | **.726±.006** | .722±.006 |
| music_style | 6.04 | **.734±.015** | .728±.009 | .706±.016 | **.731±.009** |
| mirflickr | 3.35 | .771±.026 | **.781±.046** | **.873±.002** | .871±.003 |
| image | 3 | .774±.013 | **.775±.012** | **.798±.015** | .790±.010 |
| scene | 4 | .794±.014 | **.808±.009** | .874±.012 | **.882±.008** |
| yeast | | .733±.008 | **.735±.009** | **.666±.026** | **.666±.026** |
| enron | | .665±.020 | **.670±.024** | **.668±.018** | **.668±.018** |
| corel5k | | **.440±.014** | .435±.013 | **.628±.007** | **.628±.007** |
| eurlex_dc | 10 | .797±.009 | **.800±.012** | .872±.009 | **.882±.007** |
| eurlex_sm | | **.722±.002** | .720±.010 | .827±.008 | **.831±.005** |
| delicious | | **.630±.007** | **.630±.006** | **.712±.004** | .711±.003 |
| tmc2007 | | **.820±.004** | **.820±.004** | **.886±.002** | **.886±.002** |

Table 9: Win/tie/loss counts of pairwise t-test (with $p < 0.05$) about PML-GAN vs PML-GAN′ over all 49 PML variant datasets with different numbers of candidate labels.

| Hamming loss | Ranking loss | Average precision | Macro-averaging | Total |
|---|---|---|---|---|
| 0/49/0 | 0/49/0 | 0/49/0 | 00/48/1 | 0/195/1 |

Table 10: Characteristics of the multi-label experimental datasets.

| Dataset | #Inst. | #Feature | #Classes | Domain |
|---------|--------|----------|----------|--------|
| CAL500 | 500 | 68 | 174 | music |
| emotions | 593 | 72 | 6 | music |
| image | 2000 | 294 | 5 | images |
| scene | 2407 | 294 | 6 | images |
| tmc2007 | 28596 | 49060 | 22 | text |

Table 11: Comparison of PML-GAN with the comparison methods. The best results are presented in bold font.

| Dataset | PML-GAN | PARTICLE-VLS | PARTICLE-MAP | PML-LC | PML-FP | ML-RBF |
|---------|---------|--------------|--------------|--------|--------|--------|
| Hamming loss (the smaller, the better) | | | | | | |
| CAL500 | **.143±.002** | .150±.005 | .154±.004 | .161±.005 | .161±.005 | .230±.014 |
| emotions | .228±.028 | **.226±.026** | .234±.024 | .244±.033 | .241±.033 | .323±.037 |
| image | **.122±.005** | .226±.056 | .252±.093 | .238±.048 | .261±.052 | .252±.003 |
| scene | **.130±.005** | .132±.037 | .133±.028 | .146±.031 | .161±.037 | .220±.002 |
| tmc2007 | **.079±.001** | .080±.003 | .082±.003 | **.079±.002** | .080±.002 | .109±.001 |
| Ranking loss (the smaller, the better) | | | | | | |
| CAL500 | **.190±.003** | .193±.014 | .230±.017 | .195±.009 | .195±.009 | .200±.010 |
| emotions | **.178±.013** | .232±.031 | .181±.023 | .200±.029 | .206±.025 | .236±.028 |
| image | **.087±.010** | .093±.015 | .103±.013 | .111±.020 | .128±.023 | .198±.012 |
| scene | **.082±.005** | .096±.007 | .123±.010 | .130±.015 | .152±.020 | .141±.007 |
| tmc2007 | **.045±.001** | .059±.008 | .055±.005 | .051±.006 | .051±.007 | .056±.002 |
| Average precision (the larger, the better) | | | | | | |
| CAL500 | **.499±.008** | .464±.019 | .460±.020 | .481±.016 | .480±.015 | .453±.017 |
| emotions | **.789±.018** | .764±.025 | .780±.026 | .778±.023 | .777±.030 | .770±.034 |
| image | **.848±.012** | .787±.062 | .777±.082 | .787±.086 | .755±.090 | .770±.012 |
| scene | **.855±.008** | .810±.081 | .802±.055 | .752±.059 | .717±.075 | .753±.010 |
| tmc2007 | **.826±.002** | .814±.022 | .804±.017 | .813±.016 | .814±.016 | .798±.003 |
| Macro-averaging AUC (the larger, the better) | | | | | | |
| CAL500 | .521±.014 | .519±.011 | .514±.010 | .540±.009 | **.543±.010** | .480±.012 |
| emotions | **.822±.008** | .758±.021 | .775±.030 | .810±.023 | .810±.026 | .762±.020 |
| image | **.919±.010** | .841±.131 | .806±.104 | .866±.119 | .871±.117 | .806±.008 |
| scene | **.920±.006** | .836±.045 | .831±.075 | .810±.029 | .795±.024 | .824±.045 |
| tmc2007 | **.908±.002** | .877±.007 | .881±.005 | .893±.010 | .893±.011 | .872±.004 |
| Coverage (the smaller, the better) | | | | | | |
| CAL500 | **.803±.014** | .864±.005 | .866±.007 | .809±.025 | .808±.025 | .878±.004 |
| emotions | **.315±.011** | .343±.048 | .316±.034 | .320±.033 | .332±.031 | .330±.003 |
| image | **.089±.009** | .134±.022 | .146±.020 | .211±.063 | .218±.073 | .214±.011 |
| scene | **.083±.005** | .098±.011 | .103±.015 | .100±.031 | .117±.041 | .107±.008 |
| tmc2007 | **.127±.001** | .143±.008 | .145±.008 | .140±.010 | .139±.010 | .145±.002 |
| One error (the smaller, the better) | | | | | | |
| CAL500 | .140±.036 | .156±.027 | .170±.020 | **.125±.023** | .133±.023 | .160±.021 |
| emotions | .281±.031 | **.276±.038** | .316±.034 | .339±.035 | .330±.036 | .343±.030 |
| image | **.252±.019** | .305±.094 | .319±.147 | .378±.138 | .336±.133 | .347±.018 |
| scene | **.243±.012** | .277±.012 | .275±.081 | .303±.098 | .347±.117 | .334±.017 |
| tmc2007 | **.196±.004** | .197±.031 | **.196±.026** | .209±.024 | .208±.025 | .220±.005 |

## A.6 EXTENDED EXPERIMENTS

To further validate the effectiveness of the proposed approach, we conducted an extended experiment on five datasets by adding a large portion of label noise into the true labels. Specifically, for each dataset with a total of $C$ class labels, we created the PML data in the following way: For each training instance, if it has $k$ true labels, we randomly select $\min(k, C - k)$ irrelevant labels to add and form its candidate label set. When $k \leq C/2$, the number of irrelevant noisy labels is identical to the number of relevant true labels; when $k > C/2$, the whole label set is used as the candidate label set.

In this setting, we performed experiments on five data sets with the different sizes of class labels. The number of instances, number of classes and domain for each dataset are illustrated in Table 10. We compared the proposed PML-GAN method with the five comparison methods on the five datasets. For each dataset, we randomly select 80% of the data for training and use the remaining 20% for testing. We repeat each experiment 10 times with different random partitions of the datasets. The comparison test results in terms of six commonly used evaluation metrics (hamming loss, ranking loss,

average precision, macro-averaging AUC, coverage, one error) Zhang & Zhou (2014) are reported in Table 11. We can see that the proposed PML-GAN outperforms the comparison methods in general. Specifically, among the 30 cases over 5 datasets and 6 evaluation metrics, the proposed approach outperforms PARTICLE-VLS, PARTICLE-MAP, PML-LC, PML-FP and ML-RBF in 28, 29, 28, 29, 30 cases respectively. Moreover, the performance gains yield by the proposed approach over all the other methods are quite notable. For example, in terms of average precision, the proposed approach outperforms the best alternative comparison method by 6.1% and 4.5% on image and scene respectively. These results again validated the effectiveness of the proposed approach.

