# OpenReview forum: "Adversarial Paritial Multi-label Learning"
_ICLR.cc/2020/Conference — Reject_

### Official Review · AnonReviewer1 · 2019-10-22
**Official Blind Review #1333**

**Rating:** 8

**Review:**

This paper proposed a new method for partial multi-label learning, based on the idea of generative adversarial networks. The partial multi-label learning is the problem that one instance is associated with several ground truth labels simultaneously, but we are given a superset of the ground truth labels for training. To solve the problem, the paper proposes a learning model composing of four deep neural networks, two of them for prediction the ground truth labels (the prediction network, and the disambiguation network), one of them to generate instances based on the disambiguated labels, and one discriminative network to discriminated generated instance versus true instances. The four network are concatenated, and the paper proposed to optimize the sum of generation loss, the classification loss, and the adversarial loss. Theoretical results like the theories in the original GAN paper are also given. Finally, the paper does thorough empirical studies compared the proposed method with four other baselines and three variants on fourteen datasets with various settings.

The paper is the first paper to proposes a GAN-style algorithm for the partial multi-label learning problem. The empirical results also validate its superior performance compared to other baselines on the data sets used. On the other hand, the paper fails to give enough intuition on why the GAN can naturally be used to solve the partial multi-label learning problem, and why the proposed method can perform better than other baselines. There is also some inaccuracy in discussing related work. Overall, given the contribution to introduce GAN to partial multi-label learning problem and the thorough empirical studies, I think this paper can be weakly accepted.

Main arguments
The paper argues that existing methods may not be good on this problem is due to the fact that they generally learn a “confidence score” for every candidate labels, while the learned “confidence score” may not be accurate enough, and the errors may accumulate through these process. I agree with the paper on this aspect. However, in the paper, it also learns something similar to the “confidence” score in the disambiguation network \tilde D, so why the proposed method is better than previous baselines? I guess the reason is due to the adversarial training part in the latter steps. But the paper fails to give clear intuition why the adversarial training part can lead to better performance.

I am also wondering will the proposed model leads to a trivial solution. For example, one possible solution may be that both the prediction network F and the disambiguation network \tilde D get the same solution: the given partial label set. In this way, the classification error can always be zero, and we can train the generation network to give a ground truth instance x, given that the generation network G is strong enough. So I also guess the discrimination network is the key part for the success of the proposed method empirically. So I am curious if we only use the adversarial loss without training the prediction network F, what the results will be. In the Ablation Study Sec. 5.3, the paper does not compare the results optimizing only the adversarial loss. I believe the results of such a study can help us understand what the key part of the proposed method is.

In the related work part, the arguments on “weak label learning” cannot be used for partial multi-label learning is not accurate. In fact, “weak label learning” studies the problem when positive labels are missing, and partial multi-label learning studies the problem when negative labels are missing. So in general, the methods for “weak label learning” can be used for “partial multi-label learning” if we exchange the role of positive labels and negative labels in both problems. So why we need to study partial multi-label learning? I think the reason is that, generally, in multi-label learning, the number of negative labels will be much larger than the number of positive labels. So “partial multi-label” is actually more strong supervision information than “weak labels” because positive labels are more important. Such arguments and related empirical studies (when the number of negative labels is much larger than positive labels) are suggested to be added into the further version of the paper.

Although the paper has some deficiencies, it does a good job of introducing GAN into partial multi-label learning, and it is also the first paper to use the powerful deep neural networks into this problem, which may trigger some interesting studies given the booming of neural networks these days. And thorough empirical studies are done by comparing to not only baselines but also different loss parts of the proposal it owes. It is worth reading especially it may have an impact on further studies.

---------------------------------------
I am satisfied with the rebuttal and tend to increase my score. Although the paper is not perfect, it does a good job of introducing GAN into partial multi-label learning, and it is also the first paper (thus it is novel) to use the powerful deep neural networks into this problem, which may trigger some interesting studies given the booming of neural networks these days.

Actually, I think partial multi-label learning problem is difficult and it may not possible to have perfect solution nowadays. Partial multi-class learning may be easier since you know only one label is true, but partial multi-label learning is difficult since you do not know how many true labels there are. It makes no point to require a perfect solution for such a dirty problem.

**Experience Assessment:**

I have published in this field for several years.

**Review Assessment: Checking Correctness Of Derivations And Theory:**

I assessed the sensibility of the derivations and theory.

**Review Assessment: Checking Correctness Of Experiments:**

I carefully checked the experiments.

**Review Assessment: Thoroughness In Paper Reading:**

I read the paper thoroughly.

---

### Official Review · AnonReviewer3 · 2019-10-23
**Official Blind Review #3**

**Rating:** 3

**Review:**

The authors propose a novel GAN-based method to tackle PML problem. By using Encoder-Decoder architecture, the authors combine four neural networks including disambiguator, predictor, generator, and discriminator to implement the integrated model and get an overall good performance in various experiment settings and datasets.

Pros:
- It's novel to introduce GAN to solve the PML problem.
-The model works better than other state-of-art models overall.

Cons:
- The novelty of this paper is limited to some extent. It seems that the model just combines ideas of GAN and PML.
- I wonder if GAN still works in this architecture in some situations. Under the condition when the dimension of labels is small enough, it will be hard to generate good samples because the information which can be utilized for the generator mightn't be sufficient. However, when the dimension of labels is large, there will be some combinations of labels that aren't in the training set, which may harm the performance of the generator.
- Because the relations between instances and labels aren't one-to-one in most cases, I wonder whether the five-layer perceptron still works well as a generator if one setting of labels can correspond to various kinds of instances.
- It will be better to compare the generator part proposed in this paper with a simple interpolation method in the process of extending the dataset.

**Experience Assessment:**

I have published one or two papers in this area.

**Review Assessment: Checking Correctness Of Derivations And Theory:**

I carefully checked the derivations and theory.

**Review Assessment: Checking Correctness Of Experiments:**

I carefully checked the experiments.

**Review Assessment: Thoroughness In Paper Reading:**

I read the paper thoroughly.

---

### Official Review · AnonReviewer2 · 2019-10-24
**Official Blind Review #2**

**Rating:** 3

**Review:**

The motivation of this paper is to handle noise candidate multi-labels by co-training two networks. The work trains the disambiguation network, which learns to predict the probability of each class label being the additive irrelevant label and then is used to get the disambiguated label confidence vector, and the prediction network, which learns to predict the probability of the disambiguated labels.
The additional adversarial loss and generation loss is aimed to enhance the label disambiguation by learning the mapping from the disambiguated labels to the input features.

Pros:
By minimizing the disagreement between the confidence of being the disambiguated label and the predicted probability of each label, the partial multi-label learning gets better results.

The experiments in this paper are complete and thorough. The authors have tested the model in many datasets and designed the ablation study to verify the effect of each loss. And the proposed model achieved the state-of-art results.

Cons:
However, the effect of the adversarial loss and the generation loss is doubtful because:
1) The mapping from the labels to the input features is hard to learn since the label space does not contain the complete information of input features. It is doubtful that the generation is helpful.

2) The results of the ablation study do not show consistently significant improvements.

3) In the appendix, the variant of PML-GAN, which considers an auxiliary classification loss on the generated data, has little improvement compared to PML-GAN. The author claims that it is because of sufficient training data. What if considering the insufficient training data and testing the variant of PML-GAN? I think it can somehow verify the effect of the generation.

About the writing of this paper, the motivation of the work is not clearly defined. Although we can get what the work was done, we cannot get why the work did this.

**Experience Assessment:**

I have published one or two papers in this area.

**Review Assessment: Checking Correctness Of Derivations And Theory:**

I assessed the sensibility of the derivations and theory.

**Review Assessment: Checking Correctness Of Experiments:**

I carefully checked the experiments.

**Review Assessment: Thoroughness In Paper Reading:**

I read the paper thoroughly.

---

### Decision · Program_Chairs · 2019-12-19

**Decision:**

Reject

**Comment:**

The paper considers a problem of clearly practical importance: multi-label classification where the ground truth label sets are noisy, specifically they are known (or at least assumed) to be a superset of the true ground truth labels. Learning a classifier in this setting require simultaneous identification of irrelevant labels. The proposed solution is a 4-part neural architecture, wherein a multi-label classifier is composed with a disambiguation or "cleanup" network, which is used as conditioning input to a conditional GAN which learns an inverse mapping, trained via an adversarial loss and also a least squares reconstruction loss ("generation loss").

Reviews were split 2 to 1 in favour of rejection, and the discussion phase did not resolve this split, as two reviewers did not revisit their assessments. R2 and R3 were concerned about the overall novelty and degeneracy of the inverse mapping problem. R1 increased their score after the rebuttal phase as they felt their concerns were addressed in comments (regarding issues surrounding the related work, the possibility of trivial solutions, and intuition for why the adversarial objective helps), but these were not addressed in the text as no updates were made.

I agree with the authors that PML is an important problem (one that receives perhaps less attention than it should from our community), and their empirical validation seems to support that their method outperforms (marginally, in many cases) methods from the literature. While the ablation study offers preliminary evidence that the inverse mapping is responsible for some of the gains, there are a lot of moving parts here and the authors haven't done a great job of motivating why this should help, or investigating why it in fact does. Based on the scores and my own reading of the paper, I'd recommend rejection at this time.

My own comments for the authors: I'd urge efforts to clarify the motivation for learning the inverse mapping, in particular adversarially (rather than just with the generation loss) in the text of the paper as you have in your rebuttals, and to improve the notation (the use of both D-tilde and D is confusing, and the omega notation seems unnecessary). I'm also not entirely clear whether the generator is stochastic or not, as the notation doesn't mention a randomly sampled latent variable (the traditional "z" here is a conditioning vector). Either way, the answer should be made more explicit.